# Tailoring Chemometric Models on Blood-Derived Cultures Secretome to Assess Personalized Cancer Risk Score

**DOI:** 10.3390/cancers12061362

**Published:** 2020-05-26

**Authors:** Maria Laura Coluccio, Francesco Gentile, Ivan Presta, Giuseppe Donato, Nicola Coppedè, Immanuel Valprapuram, Chiara Mignogna, Annamaria Lavecchia, Federica Figuccia, Virginia M. Garo, Enzo Di Fabrizio, Patrizio Candeloro, Giuseppe Viglietto, Natalia Malara

**Affiliations:** 1Department of Experimental and Clinical Medicine, University Magna Graecia, 88100 Catanzaro, Italy; coluccio@unicz.it (M.L.C.); immanuelloyola@gmail.com (I.V.); figucciafederica@libero.it (F.F.); virginiam.garo@gmail.com (V.M.G.); patrizio.candeloro@unicz.it (P.C.); Viglietto@unicz.it (G.V.); 2Department of Electrical Engineering and Information Technology, University Federico II, 80125 Naples, Italy; francesco.gentile77@gmail.com; 3Department of Health Science, University Magna Graecia, 88100 Catanzaro, Italy; presta@unicz.it (I.P.); gdonato@unicz.it (G.D.); mignogna@unicz.it (C.M.); 4Institute of Materials for Electronics and Magnetism, IMEM CNR Parco Area delle Scienze, 43124 Parma, Italy; nicola.coppede@gmail.com; 5Hospital Pugliese Ciaccio,88100 Catanzaro, Italy; amlavecchia@libero.it; 6Department of Applied Science and Technology, Polytechnic University of Turin, 10129 Turin, Italy; enzo.difabrizio@polito.it

**Keywords:** blood-derived cultures, cultured circulating tumor cells, secretome, cancer risk-score, chemometric model, super-hydrophobic surfaces, conductive polymer, PEDOT:PSS methylglyoxal adducts, liquid biopsy

## Abstract

The molecular protonation profiles obtained by means of an organic electrochemical transistor, which is used for analysis of molecular products released by blood-derived cultures, contain a large amount of information The transistor is based on the conductive polymer PEDOT:PSS comprising super hydrophobic SU8 pillars positioned on the substrate to form a non-periodic square lattice to measure the state of protonation on secretomes derived from liquid biopsies. In the extracellular space of cultured cells, the number of glycation products increase, driven both by a glycolysis metabolism and by a compromised function of the glutathione redox system. Glycation products are a consequence of the interaction of the reactive aldehydes and side glycolytic products with other molecules. As a result, the amount of the glycation products reflects the anti-oxidative cellular reserves, counteracting the reactive aldehyde production of which both the secretome protonation profile and cancer risk are related. The protonation profiles can be profitably exploited through the use of mathematical techniques and multivariate statistics. This study provides a novel chemometric approach for molecular analysis of protonation and discusses the possibility of constructing a predictive cancer risk model based on the exploration of data collected by conventional analysis techniques and novel nanotechnological devices.

## 1. Introduction

Among the multiple protocols investigated for the isolation and characterization of circulating tumor cells (CTCs), the methodology that isolates and characterizes them starting from a primary culture derived from peripheral blood (BDCs) has the advantage of preserving their heterogeneous composition [1,2,3,4,5]. In previous works, Malara and co-workers described a protocol, which, starting from a gradient phase, recognized a working density phase enriched for circulating tumor cells [1,2,3,4,5,6,7], and successively expanded them in a short-term culture, revealing different subsets of CTCs. The heterogeneous composition of the CTCs, cultured inside BDCs, was previously proven by tumorigenicity test [2] and by molecular features preservation between cancer cells expanded through BDCs, and cancer cells isolated in tumor tissue biopsy in the same patient [3,6,7,8]. The analysis of the secreted products released in the conditioned medium during the BDCs’ life span revealed an increased concentration of methylglyoxal adducts (MGAs) in cancer with respect to healthy secretomes. The increased levels of MGA were also analyzed with respect to the stage of tumor disease, highlighting a worse outcome [1,9]. MGAs are produced during glycolysis, lipid peroxidation, and metabolism of acetone, glycerol, and threonine [1,10]. These compounds result from chemical spontaneous interaction between methylglyoxal and glyoxal. Methylglyoxal and glyoxal are originated prevalently from glycolysis during cell metabolism by means of a strongly electrophilic process and characterized by a high kinetic rate. Free methylglyoxal and glyoxal react with nucleophilic sites of proteins, lipids, and nucleic acids, which induces the formation of final glycation adducts [1,11]. On the basis of the speed of formation and the degree of reversibility, different time phases were identified in the process, yielding, as a result, to a time classification of the adducts (i.e., early stage and advanced stage adducts). The early stage [1,12] adducts resulted from a direct relationship between glucose and amino groups of side chains of lysine residues and amino acid residues (FL). Early stage adducts are instable because they are subject to a very quick reversible reaction. The advanced stage MGAs are stable and formed by the interaction between endogenous metabolites of a-oxoaldehyde, methylglyoxal, and glyoxal, as a minor contribution with different specific distributions of the site on proteins. Advanced adducts are hydroimidazolone derived from methylglyoxal, N-(5-hydro-5-methyl-4-imidazolon-2yl)-ornithine (MG-H1), (N2)-G-H1 (G-H1), N-(1-carboxymethyl) lysine (CML), and (H8)-GOLD (GOLD) [1,12]. The aldehydes formed during glycolysis, in a normal cell, are catabolized by the glutathione system [10]. The glycation products are released in the extracellular space as a function of (i) the type of cellular metabolism, and, as importantly, (ii) the damage of the intracellular anti oxidative system. When the MGAs are present in high concentrations in the secretome of a cell culture, this indicates that the cultured cells are driven both by a glycolysis metabolism and by a compromised function of the glutathione redox system, which is unable to counteract the growing formation of reactive aldehydes. In this scenario, the analysis of the molecular protonation state (PS) of secretome, as an indicator of the conductivity state of proteins, denounces the high concentration of strongly electrophilic compounds able to interact with titratable protein side chains silencing their conductivity. On this basis, we used an organic electrochemical transistors device [13,14] based on the conductive polymer PEDOT:PSS. The device comprises super-hydrophobic SU8 pillars positioned on the substrate to form a non-periodic square lattice to measure the protonation state of secretome derived from liquid biopsy. The entire system is coated with PEDOT:PSS polymer and by a fluorocarbon polymer, which assures the hydrophobicity of the device. A solution on a similar device maintains a spherical shape as suspended in air, and specific species can be clustered on the basis of their physical characteristics. A high grade of PS differentiation is strictly connected to the high grade of damage existing in the oxidative system providing a criterion to identify cancer: healthy, and an intermediate class of subjects that are likely develop cancer [1,9]. The intermediate subset of samples corresponds clinically to healthy subjects without evidence of disease able to develop disease with a percentage of 50%, as previously reported [1]. Hereby the use of chemometric approaches, such as the exploration of multivariate data and the classification strategies, aims to build predictive cancer models based on quantitative (calibration) and qualitative (classification) evaluations on protonation profiles.

## 2. Material and Methods

### 2.1. Experimental Model and Subjects Details 

From July 2013 to December 2019, blood samples were collected prospectively from patients at the clinics of the Department of Clinical and Experimental Medicine and the Department of Medical and Surgical Sciences, University “Magna Græcia” of Catanzaro. The study named CHARACTEX (Characterization of circulating tumor cells and expansion) was approved by the local institutional reviewer board and was conducted according to the recommendations of the Declaration of Helsinki and its amendments. The study approval number is 2013.34. Written informed consent document (ICD), comprising a patient information sheet, was obtained by all enrolled volunteers. Peripheral blood samples (total volume of 5 mL) were drawn from both controls (volunteer healthy subjects) and untreated patients with a primary diagnosis of cancer, placed into tubes containing EDTA as anticoagulant. All subjects enrolled in this study, had “normal” glucose levels of 70 to 85 mg/dL and family history without cases of diabetes and/or neurodegenerative diseases. Subsequently, the samples were processed following the procedure described previously by Malara et al. [1,2,3,5,7,8], Simone et al. [4] and Guadagno et al. [6]. Patient age, sex, diagnosis, and clinical status are described in Appendix A.

### 2.2. Liquid Biopsy Primary Culture 

To develop the culture, cells of interest were isolated by working range as previously reported by Malara et al. [1,2,3,5,7,8]. After washing in phosphate buffered saline (PBS), the cells were recovered in a medium promoting in vitro expansion for 14 days (short-term cultivation). The medium composition was reported in Malara et al. [1,2,3,5,7,8].

### 2.3. Cytopathology Characterization of Liquid Biopsy 

To perform a morphological evaluation using hematossilin and eosin (H&E) staining, parallel to the cytometric evaluation, blood-derived cell cultures were established and maintained on both plates and slides. After 14 days of cultivations and regular collection of the conditioned medium, cells on slides were fixed with a 4% paraformaldehyde solution and stored at 4 °C. Successively, slides were stained following a standard H&E staining protocol [1,6,8] and evaluated under a light microscope (Leica ICC50HD, Leica Microsystem, Milan, Italy). Two pathologists experienced in cancer disease interpreted each case independently before arriving at a consensus reference diagnosis, and a cytopathological score was assigned.

### 2.4. Secretome Collection and Characterization 

The cell cultures were monitored at 48 h intervals, and each time 10% of total volume of the culture medium was collected and replaced with fresh medium. The 10% of collected medium was placed in a cuvette and stored at 4 °C. After a two-week incubation period, the culture was harvested and the medium was separated from the cellular elements by centrifugation at 1870 rpm for 15 min. The supernatant was added to the previous collected medium, filtered, and stored at −80 °C. The pellet formed was collected and used for successive characterizations [1].

### 2.5. Immunoblotting 

Total protein content of the extracts was determined using the Bradford protein assay (Bio-Rad, Segrate (MI), Italy) with bovine serum albumin (Sigma Aldrich, Milano, Italy) as standards. Fifty micrograms of each protein extract were diluted with Laemmli buffer and incubated at 100 °C for 5 min. Subsequently, all samples were loaded onto a 12% SDS-polyacrylamide gel and electrophoresed at 80 V. After electrophoretic separation, proteins were transferred to a nitrocellulose membrane using a Trans-Blot Turbo (Bio-Rad, Segrate (MI), Italy) protein transfer system. Membranes were first blocked in 5% nonfat milk / 1X TBS-T and then incubated over night at 4 °C, with a 1:500 dilution in the same blocking buffer, of mouse anti- methylglyoxal-protein adducts. After washes, membranes were incubated with 1:2000 dilution of HRP-conjugated anti-mouse IgG (Cell Signaling Technology, Pero (MI), Italy). Immunocomplexes were detected by using the SuperSignal West Femto ECL substrate (Pierce, Pero (MI), Italy). Fully automated densitometric software, Alliance 2.7 1D (UVITEC, Eppendorf, Milan, Italy), was used for acquisition and analysis of immunoblots images. 

### 2.6. Raman Spectroscopy 

Secretomes from primary cultures derived from peripheral blood of healthy (HS) and cancers subjects (CS) were lyophilized and analyzed in the solid state on a CaF_2_ substrate by an InVia Raman microscope from Renishaw Ltd., working in backscattering configuration with an 830 nm laser source, focused on the sample with a 50×/0.75 NA objective. The Raman spectra were collected in the spectral range from 800 to 1800 cm^−1^ using a power of 65 mW and an integration time of 50 s. Signal was analyzed with a 1800 lines/mm grating. All the Raman spectra were baseline corrected using four- to five-order polynomials to eliminate the auto-fluorescence background.

### 2.7. Fabrication of the Device 

The SeOCET was given by the superposition and correct alignment of different layers, as described in [13,14,15,16,17,18]. Layer A is a silicon substrate that contains conductive gold circuits connecting the electric active areas in the interior of the device to the metal contacts (source and drain electrodes) positioned at the border of the device for experimental convenience. Layer B, which comprises super-hydrophobic SU8 micro pillars, was arranged to form a square pattern in which the spacing of the pillars was not constant over the domain, which permitted the automatic overlap of the solution droplet with the center of the device. Five pillars positioned at the center of the pattern were equipped with micro-electrodes; the final device contained active spots that smoothly transitioned from the center (sensor number 3) to the border (sensor numbers 1 and 5) of the droplet. In doing so, the device could measure the electric activity of active species in specific points of the solution, and this could be space-resolved. The entire substrate was therefore spin coated with a conductive PEDOT:PSS thin film. A fluorocarbon polymer (C4F8) was finally deposited on the devices on account of their hierarchical structures bridging differently.

#### 2.7.1. Microfabrication of Gold Conductive Patterns on the Supporting Silicon Substrates (Layer A) 

P-doped, (100) silicon wafers with resistivity of 5–10 Ohm/cm were used as a substrate. They were cleaned with acetone and isopropanol to remove possible contaminants and then etched with a 4% hydrofluoric acid (HF) solution. The wafers were then rinsed with DI water and dried with N_2. Standard optical lithography techniques were used to generate patterns of the circuits within a layer of positive tone resist (S1813). The masks necessary for optical lithography were fabricated using direct EBL (electron beam lithography). Then, a 70 nm thick layer of gold was deposited on the sample. Conventional lift-off process in an excess of acetone was used to remove the un-exposed resist from the substrate, thus defining the pattern of gold circuits on the substrate.

#### 2.7.2. Microfabrication of Patterned Super-Hydrophobic Surfaces (Layer B)

Non-periodic patterns of super-hydrophobic micro pillars were fabricated on the substrate. The lattice of the pillar was a non-periodic lattice where an increasingly higher density of pillars was found moving from the periphery to the center of the pattern [16,17]. At the border of the pattern, the distance between adjacent pillars was δ = 20 μm, and at the center of the pattern, the distance was δ = 2 μm. The distance (δ) between different pillars could be expressed as a function of the position (r) of those pillars from the center of the device as δ = (2 + 0.9 δ_o_ (r⁄l)^0.35^) μm, where l = 400 μm is the extension of the patterned device, and δ_o_ =20 μm is a constant. The diameter of the pillars is d = 10 μm while their height is h = 20 μm. The non-periodic tiling of the surface results in a non-uniform surface energy distribution, with a minimum of energy at the center of the pattern that assures automatic positioning of the drop on the pattern. P-doped, (100) silicon wafers with resistivity of 5–10 Ohm/cm were used as substrates; they were cleaned with acetone and isopropanol to remove possible contaminants and then etched with 4% HF solution. The wafers were then rinsed with DI water and dried with N_2. Standard optical lithography was used to generate the pattern of pillars using a negative-tone resist SU8-25. The masks necessary for optical lithography were fabricated using direct EBL (electron beam lithography). The correct alignment of layer A with layer B was assured by alignment markers conveniently positioned at the margins of both patterns.

#### 2.7.3. Realizing Micro-Electrodes on the Top of the Pillars 

Some pillars were further modified to incorporate metallic contacts, which connected the described pillars to the gold contact area. This task was performed using EBID (electron beam-induced deposition) fabrication. This process consists of injecting a precursor gas including Pt–C into the SEM chamber; the gaseous molecules are then hit by the electron beam and precipitate onto the substrate with a high spatial accuracy. For each pillar, the platinum deposition was conducted in three steps. In the first step, the contacts were deposited on top of the pillars. An SEM column current of 1.6 nA and voltage 20 kV were set, and these permitted deposition of a layer of platinum approximately 100 nm thick. In doing so, we fabricated horizontal lines, which protruded from the pillar, creating a deposition on the bottom useful for a later connection to the gold structure. The second step consisted of the connection to the gold track. Since the distance between the lines ranged from 60 μm to 70 μm microns, the process of fabrication was considerably time consuming. To overcome this limitation, we increased the electronic current and voltage to 6.4 nA and 30 kV, respectively. In the last step, we realized the connection on the side of the pillar. To do so, we turned and tilted the substrate to an angle of 45° in order to deposit on the lateral surface of the pillar.

#### 2.7.4. The Conductive PEDOT:PSS Thin Film

A solution of poly(3,4-ethylenedioxythiophene) doped with poly(styrene sulfonate) (PEDOT:PSS) (H.C. StarckClevios PH500), was then spun onto the silicon substrate with the pillars yielding a thickness of h∼80 nm. The PEDOT:PSS solution was previously doped with ethylene glycol (Sigma Aldrich), to enhance its electrical conductivity, and dodecyl benzene sulfonic acid (DBSA) surfactant (Sigma Aldrich), to improve film forming. After spinning, the device was baked on a hotplate at 140 °C for 60 min. The substrates were then covered with a thin (few nm) film of a Teflon-like (C4F8) polymer to ensure hydrophobicity; to do this, a modified Bosch/RIE process was utilized, with the sole passivation mode activated. In this phase, all gas flows, including SF_6_, argon, and oxygen, were set to zero, with the exception of the chemically inert passivation layer C_4_F_8_.

### 2.8. SeOCET Operation

Samples containing cell culture liquids were gently positioned upon the active surface of the biosensor device in the form of drops of volume V < 10 μL. The electrical response of biosensors was measured using a two-channel source/measure precision unit (Agilent B2902A), controlled by homemade software. Biosensor measurements were acquired by measuring the drain current I_ds versus time under a constant drain voltage Vds = −0.1 V. In the process, the voltage at the gate Vgs was varied between 0 and 1 V with increments of 0.2 V over a time interval of 120 s. The current response of the biosensor was expressed as current modulation ΔI/I_0 = (Ids – I0)/I0, where Ids is the drain current value measured for Vgs > 0 V, and I0 is the Ids value at Vgs = 0 V. The time constant was the time necessary to the output to reach the 63% of its final value, which was determined by fitting of the output with an exponential function.

### 2.9. Statistical Analysis 

A three way analysis of variance (ANOVA) was used to explore relationship among marker expressions, clinical parameters, and tumor characteristics. The significance level was set at *p* < 0.05. Comparison between patients and control group was performed using Mann–Whitney and Kolmogorov–Smirnov tests with a valid statistical significance of *p* < 0.05. Sub-groups were compared using the *t*-test (for continuous variable) and chi-square test or Fisher test (for categorical variables). All statistical analyses were performed using MedCalc for Windows, (MedCalc Software bv, Ostend, Belgium; https://www.medcalc.org; 2019). For hierarchical clustering, pairwise distances among clinical parameters, such as age, sex, and status within the control group and stage, grade, sex, and age within the patient group, were calculated using correlation coefficient and Euclidean distance. Dendrograms were generated using Ward’s method.

## 3. Results

### 3.1. Protonation Profiles

Secretomes from healthy and cancers subjects were also analyzed by SeOCET. This device is able to discriminate samples with different physical/chemical characteristics exploiting the super-hydrophobicity of the array of micro-pillars and at the same time, the presence of nanoelectrodes on some of them (Figure 1a). The non-periodic tiling of the micro-pillar arrays was reflected by a non-uniform surface energy density, which generated in turn a system of radial forces that recalled the drop to the center of the lattice for automatic sample positioning. Some of the pillars (nanosensors) were individually contacted to an external electrical probe station for site selective measurement on the sample surface; they incorporated nano-gold contacts with sub-micron reciprocal distance that generated enhanced and localized electric fields. They were placed symmetrically respect to the center of the device. When a drop of secretome was positioned on the device, a voltage between 0 and 1 volt was applied, generating a current of ion Ids, which flowed from the sample through the circuit and was measured by the nano-sensors. Ions in the samples were subjected to the combined effect of convective Marangoni flows and electric field and, consequently, migrated separately in relation to their size, charge, and diffusion coefficient. Most of the molecules moved towards the sides of the drop, and only a small amount of the largest molecules remained in the center where the drop was in contact with the device (Figure 1b,c). The Ids variation with voltage was measured as a function of time and could be modelled as the output of a first order system; the signal smoothly transitioned from a minimum to a steady state value (Figure 1d). The modulation m is the difference between the final and the initial value of current, normalized to one; it is a measure of the strength of the signal. The time constant τ is the time necessary to the device to reach the 63% of its final value; it is a measure of the rapidity of the signal (Figure 1d). Each of the signals could be described by the sole m and τ. Ion distribution was monitored in space and time by the arrays of sensors distributed on the device, and a phase space diagram was used to represents the state of the system at a specific time, for a specific sensor, and for a given value of external voltage V in the m–τ plane (Figure 1d, sensor S1) [1]. The different location of samples in the diagram corresponds to their different characteristics. Secretomes from cancer subjects had an altered protonation state due to the protonation and deprotonation reactions, which involve the set of titratable groups of the protein not implicated in the structural peptide bonds. SeOECTS can evidence the different protein protonation state (PPS) of samples through a phase diagram representing data from healthy, cancer, and intermediate subjects [1]. In Figure 1e the phase space diagrams related to S2, S4, and S5 sensors indicated as points representative of healthy and cancer subjects tended, in particular, to cluster for peripheral sensors (S1 and S5), corresponding to the points with higher molecular density. It is also possible to distinguish in Figure 1e the intermediate cases, localized at approximately the halfway point.

### 3.2. Blood Derived Cultures of Enrolled Volunteers

The prospective project, Characterization of circulating tumor cells and expansion (CHARACTEX), enrolled (i) oncologic patients, to monitor them through circulating tumor cell (CTCs) detection, making assessments relevant to treatment planning and to the monitoring of cancer resistance, and (ii) control subjects for early cancer detection. Blood derived cell cultures were performed from peripheral blood of 50 voluntary oncologic patients (median age 55 yr.; 75% female and 24% male; 46% I–II and 54% II–IV stage) and 50 voluntary healthy subjects (median age 54 yr.; 15% female and 84% male, 71% healthy subjects, 17% with inflammation and 11% with previous cancer). Characteristic baselines of voluntaries enrolled are detailed in Appendix A. The protocol aimed to reduce hematological contamination and enriching for cells of tissue origin (Figure 2a) [2], including short-time primary cultures both in the chamber-slide and in plates (Figure 2b,c). The first type of culture was used for cytological evaluation; the second was used for proliferating phase assessments as described in [2,3,4,5]. All cancer patients involved in the CHARACTEX project developed BDCs in both system of culture and were enriched for cancer cells independently from the type and stage of disease. The analysis of the cytological elements found in the cytological preparations obtained from cancer patients correlated with the cells presented in corresponding histopathological preparations, as previously described [6,7,8].

### 3.3. Quantitative Evaluations of Proliferation Rate in BDCs

The plate-cultured cells were analyzed for the distribution of the phases of the cell cycle and for quantifying the proliferative phase S. The average of the percentage of the phase found in the blood-derived cultures of subjects belonging to the control group was 17% ± 0.6 and 57% ± 0.2 was the average the percentage of the cultured cells in S phase found in BDCs of cancer patients (Figure 3a). Comparative evaluations showed a high and significant statistical difference between the two groups of subjects. Cumulative frequency distribution of percentage of S phase in the control group showed (Figure 3b) S phase percentage value ranging between 0% and 35%. Based on molecular protonation measurement on secretome produced by the BDCs performed form blood samples of the healthy group, two subtype or classes defined healthy control and intermediate were identified. As shown in Figure 3c, within the healthy group the percentage of S phase characterizing the BDCs belonging to intermediate class ranging between 25% and 35% ± 0.8 showed a significant difference with respect to the control-healthy class (*p* ≤ 0.0001).

### 3.4. Qualitative Evaluations of Cytology Landscape in BDCs

Parallel cultivation on chamber-slides was performed and evaluated by a scoring system (Figure 4a,d). Pathological variables considered were the rate of lympho-monocyte elements (Vc1) and endothelial cells (Vc2) (Figure 4a), structural abnormality on cells (atypical cells) (Vc3), mitosis figures (Vc4), and cluster cell formation (Vc5) (Figure 4b). Individual scores for each variable were estimated following scoring parameters (i.e., 1–5 atypical cells/100 cells corresponding score 1; 5–10 atypical cells/100 cells corresponding score 2; >10 atypical cells/100 cells corresponding score 3). Hierarchical clustering was performed to classify each slide according to compositional similarity (Figure 4c,d). The compositional pattern in control group was prevalently of lympho-monocytes elements (Vc1) and endothelial cells (Vc2). The analysis of the variability across and within the healthy group revealed subtle differences in endothelial cells (Figure 4a). In the second group (cancer patients), each stage of cancer disease displayed compositional similarity (Figure 4d). In the first and second stage of disease, the prevalent pattern combined lympho-monocytes elements (Vc1), endothelial cells (Vc2), and atypical cells (Vc3). In the late (third and fourth) stage, atypical cells (Vc3), cluster cell formations (Vc4), and mitotic figures (Vc5) prevailed (Figure 1e). Significant correlation between the number of atypical cells and the advancement of disease (Spearman’s coefficient of rank correlation rho, r = 0.5) was found and also for mitotic figures (r = 0.3 and 0.6 respectively) and cluster cell formations (r= 0.3 and r = 0.5, respectively). No correlations with patient’s age, sex of the patients, or the pathological variables considered were found.

### 3.5. Quantitative Evaluations of MAGs on BDC Secretome

Secretomes were analyzed for the presence of advanced glycation products. Immunoblotting and normalizing intensity band levels showed a different distribution of concentration of MGAs in the secretomes collected form BDCs of volunteers enrolled (Figure 5a,d). The median value of MGA was 0.6 ng/mL for the secretome produced by BDCs of the healthy group, ranging from 0.3 to 1.2 ng/mL on 10^5^ cells, and between 1.2 and 2 ng/mL on 10^5^ cells for secretomes produced by BDCs of the cancer patients group. Moreover, considering the median value of 57% for the percentage of cultured cells in S phase, comparative evaluation between BDCs of cancer patients group, characterized by a high percentage of S phase (>57%) and low (<57%), showed a significant difference (*p* = 0.04) (Figure 3c). On the other hand, in BDCs of healthy group, the median percentage of proliferating cells was 17%, and the comparative evaluation between low and high proliferation rate was no significant (Figure 3d). Healthy samples displaying an S phase >17% were characterized by protonation state measurement as intermediate, and by a concertation of MGAs within the secretome produced a 0.9 ng/mL median value. Statistical comparative analysis through independent sample *t*-test confirmed a significant difference between cancer patients and healthy group, and within the healthy group, significant difference was found between the two classes of healthy-control and intermediate (*p* = <0.0001).

### 3.6. Qualitative Evaluations by Using Spectroscopy RAMAN on BDC’s Secretome

Raman analysis performed on cancer patients and healthy voluntaries showed spectra very similar to each other due to the presence of very complex mixtures. In Table 1 are reported some principal peaks distinguished and assigned on the bases of data available in literature. Nevertheless, the average spectra relative to cancer patients (Tumor) and healthy subjects (Healthy) evidence a difference in the peak at 940 cm^–1^ assignable to SH of cysteine (Raman spectra in Figure 6a). Cancer patients (Tumor) spectra showed a minor intensity in that peak respect to the healthy subjects and the high presence of methylglyoxal can give an explanation to that different behavior between the two groups because the adducts formed by methylglyoxal and proteins occur also with cysteine blocking the thiol groups, as the sketch in Figure 6a shows. Comparing the average Raman spectrum of subjects previously defined as intermediate with that of Healthy and Tumor spectra, it is possible to notice a behavior near to Tumor spectra with a relevant decrease of the SH peak (Figure 6b).

## 4. Discussion

The chemometry offers a range of techniques both for the exploratory analysis of multivariate data and for the construction of preventive strategies. Exploratory analysis is the first step in the processing of chemometric data. This approach is aimed at providing an impartial image of the distribution of data by summarizing their main characteristics. Exploratory analysis on secretoma samples has been preliminarily performed by using a surface enhanced organic electrochemical (SE-OECT) device. The analysis enabled to identify distinctive characteristics of healthy and oncological samples, related to the protonation state of the molecules secreted (i.e., the secretome) by BDCs 14 days aged. The spatial and biochemical composition of the secretoma was strictly conditioned by the type of isolated cells. BDCs were spontaneously enriched for cancer cells for their proliferation features in cancer patients’ respect to healthy subjects. This different composition in the type of short-time cultivated cells between the two categories of volunteers is the cause of the differences emerged with the secretome analysis on the device. Then, upon a preliminary analysis of data with the SeOECT, biological data were categorized using either (a) discriminative methods [23,24] and (b) class modelling methods [25]. The first method is based on the identification of distinctive variables for each category. The second method is based on the identification of similarities between individuals belonging to the same category [26]. The study presented here bases the discriminative chemometric approach on the physical properties of the secretome, analyzed using the SeOECT device. The physical properties of secretomes are conditioned by the molecular gradient of MGAs produced by metabolic switch and relative to the oxidative stress. The analysis is based on the assumption that the detection of MGA is a marker of the degree of oxidative failure and can be used to discriminate, two main profiles for MGA levels within the healthy group.

In order of MGAs levels corresponding to distinct PS values, the healthy group were dichotomized into two subgroups or classes: the healthy-control and the intermediate classes. As previously reported, the intermediate class will develop cancer in 50% of cases within 48 months from the onset of the pathology [1]. The identikit of intermediate BDCs has also been defined here and the finger printing properties shared by the samples belonging to this category are resumed in Figure 7. Figure 7 shows the matrix of the chemometric model built in function of the discriminative and similarity variables explored in this study. The BDCs belonging to the intermediate group (i) are characterized by a proliferative phase ranging between 25% and 35%, (ii) have a cytology [8,24] characterized mainly by reactive endothelial cells and (iii) their secretoma have a Raman spectrum characterized by a reduction of the thiol residues. As previously demonstrated [1], the samples of this class have a higher concentration of advanced glycation products than the healthy control class characterized by a prevalence of early glycation products. The early glycation products are chemical compounds preserving a high degree of reversibility compared to advanced forms difficult to revert [27]. The intermediate class describes that borderline condition existing between the absence of oxidative stress and its highest complication represented by the neoplastic transformation. The epigenetic enhancer or the irreversible genomic mutation targeting the conversion on this risk condition in a cancer pathology it is not always identifiable, but the molecular fingerprinting identified here, for this class of samples, allows us to define them as being at high risk of cancer. How to evaluate the reversibility margins existing to revert the intermediate in healthy condition represents an intriguing point of view. At the same time, the reversibility potential within the intermediate condition could justify a therapeutic intervention aimed at interfering or slowing down the potential subsequent cancer disease.

The identification of the subjects belonging to the class *High risk* of cancer, in function of its distinctive biochemical trait, suggests the need to activate personalized surveillance programs. The subjects at high risk of cancer could be recruited into a follow-up program, as the individual with proven genetic predisposition. The health costs of a follow-up [28] are up to 1000 times lower than the health costs that are faced in treating a cancer diagnosed in an advanced stage [29]. 

## 5. Conclusions

The chemometric matrix is a fundamental step to build a model on which to base regression studies and to qualify the predictive value of this multidisciplinary approach. Further studies will be necessary to verify whether the empirical nature of the chemometric model presented here can be influenced by a series of variables (increase in the sample size, representativeness of the sample, etc.) and to validate and calibrate it on multiplatform device to acquire the same information directly from whole blood bypassing the culture procedures.

## Figures and Tables

**Figure 1 cancers-12-01362-f001:**
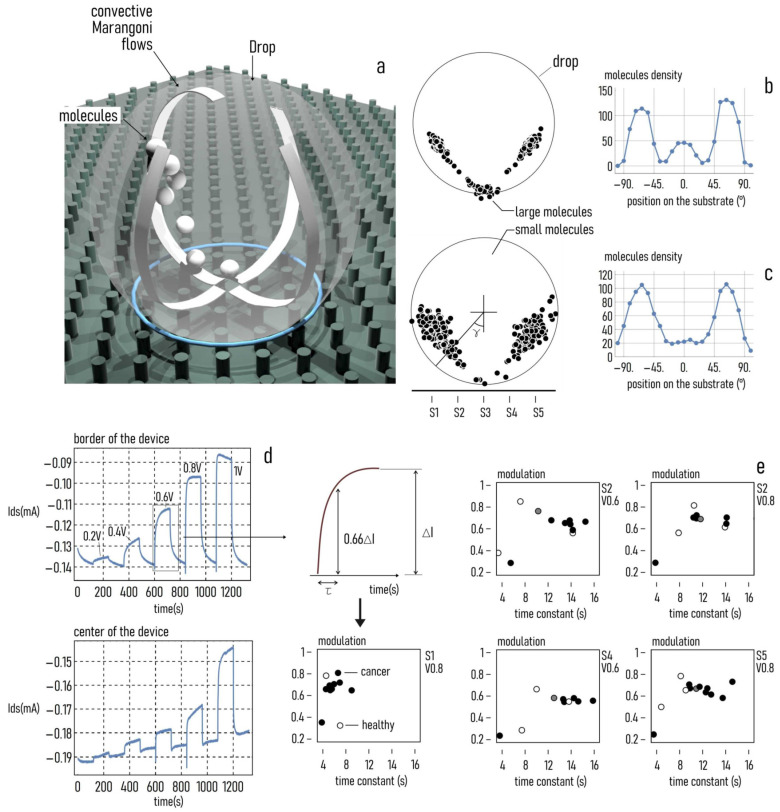
Sketch of SeOECT device. A drop of biological solution positioned on the SeOECT maintains a spherical shape and develops Marangoni convective flows within its volume (**a**). Due to the Marangoni flows, biological species dispersed in the drop are transported in different regions of the domain depending on their size; smaller molecules are preferentially transported to lateral extremes of the drop (**b**), while larger molecules preferentially accumulate in close proximity of the contact area of the drop with the substrate (**c**). Upon application of an external voltage, the device measures a continuous current of time that is indicative of the physical characteristics of the species in solution (**d**). The current of ions measured by the SeOECT may be described by the two sole parameters time constant and modulation; thus, each biological sample can be reported in a modulation vs. time constant diagram; depending on their characteristics, different samples assemble in different regions of the diagram. Convenient clustering of points in the diagram can operate accurate sample classification; hollow, black solid and grey solid circles refers respectively to healthy, cancer and intermediate subjects, (**e**).

**Figure 2 cancers-12-01362-f002:**
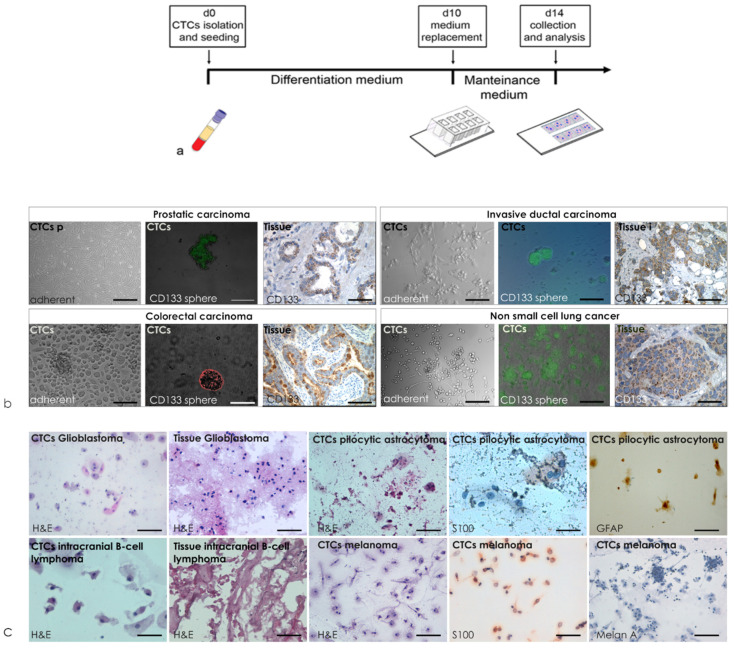
Graphic representation of the short-time blood derived cultures. (**a**) Isolation of a heterogeneous pool of circulating tumor cells (CTCs) seeded directly on slide-chambers and (**b**) on plates with a specific medium culture; (**c**) on day fourteen, the chambers were removed and the cells adherent on both were used for cytological evaluation (scale bars 10 μm).

**Figure 3 cancers-12-01362-f003:**
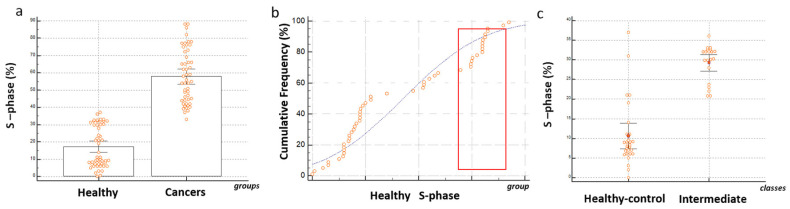
Proliferation rate in Blood-derived cultures. (**a**) Each orange dot plots correspond to individual blood-derived cultures data. Comparative independent sample t-test between the percentages of cultured cells in S phase in control and cancer blood-derived cultures. *p* ≤ 0.0001; (**b**) cumulative frequency distribution of percentage of S phase in the control group. The blue line shows the trend in observed distribution of S-phase percentage values. The orange frame indicates the range of S phase values characterizing the intermediate class individuate through protonation measurement. In (**c**) independent sample test reports a significant difference *p* ≤ 0.0001 between the two classes of samples belonging to healthy group in order of percentage of cells in S-phase at 14 days of culture.

**Figure 4 cancers-12-01362-f004:**
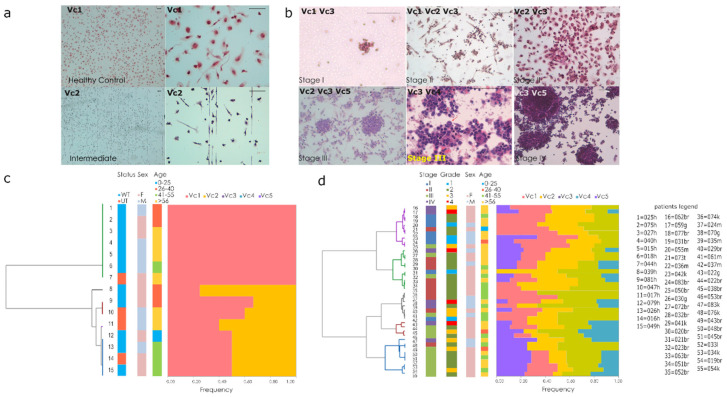
Cytology landscape in Blood-derived cultures (**a**) representative BDCs with lympho-monocytes elements (Vc1) and endothelial cells (Vc2) of control (Scale bars 100 μm). (**b**) Representative BDCs of cancer patients with atypical elements (Vc3), endothelial cells (Vc2), cluster and mitotic figures (Vc4, Vc5), i.e., in stage I a thyroid tumor case; stage II breast lobular (Vc1, Vc2, Vc3) and ductal (Vc2, Vc3) cases; stage III melanoma (Vc2, Vc3, Vc5) and NSCLC (Vc3, Vc4) cases; stage IV colon adenocarcinoma (scale bars 100 μm). (**c**,**d**) Hierarchical clustering based on Ward’s method for healthy (**c**) and cancer patient (**d**) each slides.

**Figure 5 cancers-12-01362-f005:**
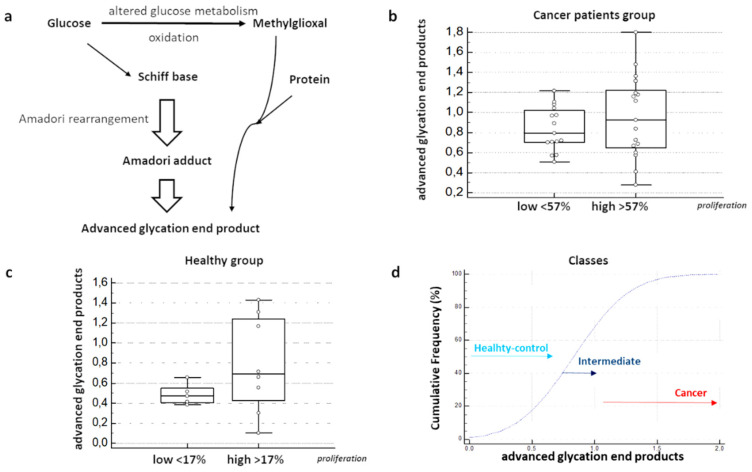
MAGs quantification on secretome of BDCs: (**a**) diagram showing metabolic pathway that underwent the production of advanced glycation products in cancer cells; (**b**) media content of MGA is distributed in order to the correspondent percentage of cells in S phase < (low) or > (high) the mean value of 57% (*p* ≤ 0.0001). In (**c**) media content of MGA is distributed in order of the correspondent percentage of cells in S phase < (low) or > (high) the mean value of 17% for healthy group (*p* ≤ 0.0001). (**d**) Comparative evaluation by using independent sample test between cancer patients and healthy groups in order of different percentage of S phase and content of MGAs.

**Figure 6 cancers-12-01362-f006:**
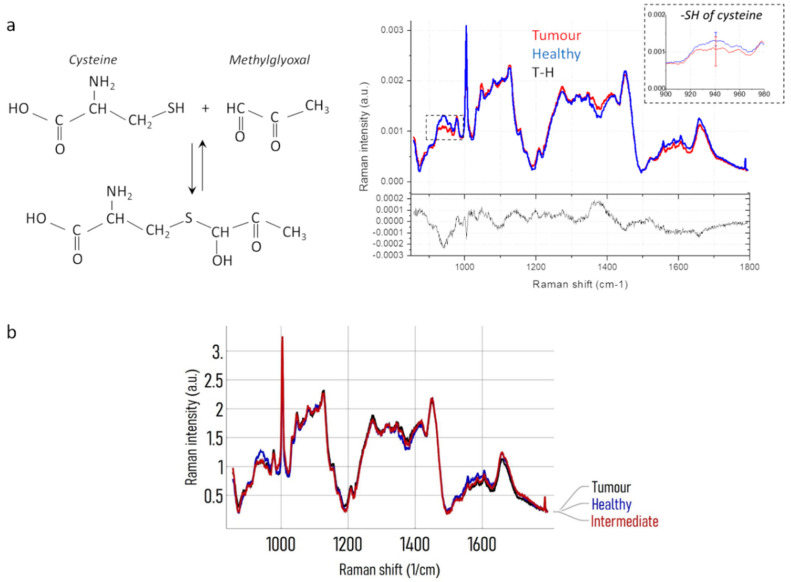
Raman Spectra of Secretome produced by Healthy and Cancer patients BDCs. (**a**) Scheme of the reaction between cysteine and methylglyoxal (left side); comparison between the average Raman spectra of cancer patients (Tumor) and healthy subjects (Healthy) and, in the bottom, their difference spectrum (right side). (**b**) Average Raman spectrum of intermediate cases, Healthy H and Tumor T spectra.

**Figure 7 cancers-12-01362-f007:**
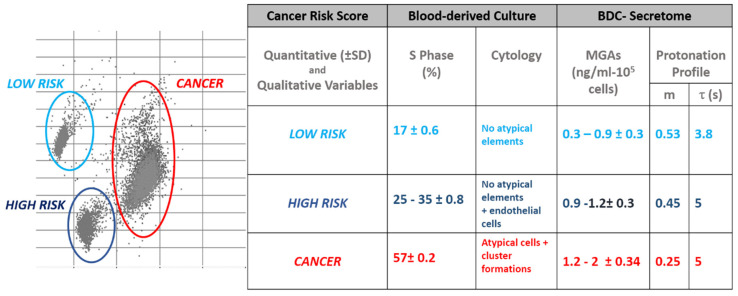
Matrix of the chemometric model. Equivalence between the experimental profiles defined as a function of quantitative and qualitative variables and exploratory classifying data of chemometric model prototype on BDCs and secretome.

**Table 1 cancers-12-01362-t001:** Principal Raman band assignments for the analyzed spectra of healthy, cancer, and intermediate subjects.

Raman band (cm^–1^)	Assignment	Reference
940	SH of cysteine, ν(CCN)symm, ν(CC) glucose	[19]
976	Phe, C–C	[19]
1004	O–P–O sym str., BK	[19]
1126	C–N str., C–C	[19]
1156	Amide III, =CH def	[19]
1208	Amide III, =CH def	[19]
1249 (shoulder)	G, amide III, CH twisting vibration	[19,20]
1314	A, G, CH vibration	[19]
1346	CH bending vibration	[21,22]
1405	CH2 def	[19]
1447	A, CH def	[22]
1520 (small)	C=C ring, methyl cytosine	[20]
1555	Phe, Tyr, C=C	[19]
1605	Amide I, C=C	[19]
1655		[19]

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
