# Peer review of "Tailoring Chemometric Models on Blood-Derived Cultures Secretome to Assess Personalized Cancer Risk Score"

_cancers, 2020, doi:10.3390/cancers12061362_

Round 1
Reviewer 1 Report
The present study reports on a novel chemometric approach applicable in the context of the molecular analysis of protonation to assess personalized cancer risk score. This is an interesting topic and the manuscript is well written.
Minor comments:
- The quality of some figures is very poor (for example Fig. 2). Please upload the figures in higher quality format.
Author Response
Paper: Manuscript ID: cancers-775627
Title: Tailoring chemometric models on blood-derived cultures secretome to assess personalized cancer risk score
Dear Editor,
We are pleased to submit to Cancers, a revised version of the manuscript entitled “Tailoring chemometric models on blood-derived cultures secretome to assess personalized cancer risk score”, and a detailed response to the comments of the Reviewers. The MS has been revised to comply with the observations of the Reviewers. Specifically:
o We have corrected typos everywhere in the MS and we improved English and style of the MS. Where necessary, we have rewritten words or sentences to convey more clearly the message of the paper.
o We have re-written the discussion.
o We have improved the quality of figures.
Considering the corrections made to the manuscript, and the positive comments of all the Reviewers on the work, we hope that the paper can be accepted for publication in this present form. We warmly thank the Reviewers that, with their comments, contributed to improve the work.
In the following, you will find the original comments from the reviewers (bold black text) and the point-by-point response of the authors (bold blue text). In the manuscript all modifications are tracked
With Regards
The authors
Review 1:
The present study reports on a novel chemometric approach applicable in the context of the molecular analysis of protonation to assess personalized cancer risk score. This is an interesting topic and the manuscript is well written.
Minor comments:
The quality of some figures is very poor (for example Fig. 2). Please upload the figures in higher quality format.
Thanks for the appreciation. As regards, the quality of figures has been modified and we hope that in their quality will meet your requests.
Reviewer 2 Report
The study ‘Tailoring chemometric models on blood-derived cultures secretome to assess personalized cancer risk score’ discusses the use of an organic electrochemical transistors device based on the conductive polymer PEDOT:PSS. This device measures the protonation state of the secretome derived from liquid biopsies of patients. The device contains super-hydrophobic SU8 pillars which are positioned on the substrate to form a non-periodic square lattice. Using this device, the authors aim to construct a predictive cancer risk model and present a thorough study of their efforts. However, some major revisions should be made before publication.
General
- The main issue is that it should be made clear why this device or way of predicting cancer risk is novel or of (financial, diagnostic, and any other) benefit to us compared to other prediction models and methods.
- The discussion in general asks for more in depth and critical analysis, argumentation and references.
- There are some significant English spelling/grammar mistakes, please look into this.
- The sentences are often long and unclear, this should be improved.
Abstract
Line 14-16 (first sentence of the abstract): this sentence doesn’t read very well (‘odd sentence’), consider changing to: “The molecular protonation profiles obtained by means of an organic electrochemical transistor, which is used for analysis of molecular products released by blood-derived cultures, contain a large amount of information.”
Line 18: ‘the liquid biopsy’ is incorrect, please change to ‘liquid biopsies’ or ‘derived from liquid biopsy’
Line 18-20: ‘Odd sentence’. Consider changing to: “In the extracellular space of cultured cells the number of glycation products increase driven both by a glycolysis metabolism and by a compromised function of the glutathione redox system.”
Line 21-23: This sentence is unclear/partially incorrect. Consider changing “On this way” to “Due to this”, “to counteracting” = incorrect time match. “Interrelating the cancer risk”, consider rephrasing for clarity.
Line 24: please change “through” to “by”
Line 24-27: This sentence doesn’t read very well and is unclear. Consider changing the first part to something like “This study provides a novel chemometric approach for molecular analysis of protonation and discusses the possibility of constructing a predictive cancer risk model based on….
Introduction
Line 31-33: ‘Odd sentence’.
Line 34: please change ‘form’ to ‘from’.
Line 44-45: incorrect English.
Line 46: ‘play the’ = incorrect verb.
Line 61: ‘to counteracting’ wrong time match.
Line 70: remove word ‘based’.
Line 70-72: consider adding more depth by commenting on literature or other studies than your own that also prove this statement.
Line 75: “aims the possibility” is incorrect.
Material and Methods
2.1
Line 78: wrong time match
Data files S1 and S2 missing?
It would be of value to describe the patient group more elaborately, particularly some information on the diagnosis/main types of cancer.
2.5
Please improve the English.
Results
3.1
Line 206-214: English could be improved.
Figure 1.b /c small drops not really visible in pdf format?
3.2
Please improve the English.
Most of this part belongs in the methods section.
3.3-3.6
Quality of the figures is poor.
Discussion
Line 361: ‘based on experimental profiles collected on samples’ incorrect English.
Line 369-371: this sentence doesn’t read very well “odd sentence”.
First part of discussion until line 392 is more suitable for the methods section.
Line 401-405: incorrect English.
Line 407-409: this is a bold statement and also different for every tumour type or country. Where is the reference?
Author Response
Paper: Manuscript ID: cancers-775627
Title: Tailoring chemometric models on blood-derived cultures secretome to assess personalized cancer risk score
Dear Editor,
We are pleased to submit to Cancers, a revised version of the manuscript entitled “Tailoring chemometric models on blood-derived cultures secretome to assess personalized cancer risk score”, and a detailed response to the comments of the Reviewers. The MS has been revised to comply with the observations of the Reviewers. Specifically:
o We have corrected typos everywhere in the MS and we improved English and style of the MS. Where necessary, we have rewritten words or sentences to convey more clearly the message of the paper.
o We have re-written the discussion.
o We have improved the quality of figures.
Considering the corrections made to the manuscript, and the positive comments of all the Reviewers on the work, we hope that the paper can be accepted for publication in this present form. We warmly thank the Reviewers that, with their comments, contributed to improve the work.
In the following, you will find the original comments from the reviewers (bold black text) and the point-by-point response of the authors (bold blue text). In the manuscript all modifications are tracked
With Regards
The authors
Review 2:
The study ‘Tailoring chemometric models on blood-derived cultures secretome to assess personalized cancer risk score’ discusses the use of an organic electrochemical transistors device based on the conductive polymer PEDOT:PSS. This device measures the protonation state of the secretome derived from liquid biopsies of patients. The device contains super-hydrophobic SU8 pillars which are positioned on the substrate to form a non-periodic square lattice. Using this device, the authors aim to construct a predictive cancer risk model and present a thorough study of their efforts. However, some major revisions should be made before publication.
General
- The main issue is that it should be made clear why this device or way of predicting cancer risk is novel or of (financial, diagnostic, and any other) benefit to us compared to other prediction models and methods.
- The discussion in general asks for more in depth and critical analysis, argumentation and references.
- There are some significant English spelling/grammar mistakes, please look into this.
- The sentences are often long and unclear, this should be improved.
We thank the reviewer for the constructive criticisms aimed to improve the MS. We edited the discussion as requested and discussed on the way of predicting cancer risk is novel or of (financial, diagnostic, and any other) benefit. Moreover, we have improved English and style of the MS and where necessary rewritten words or sentences to clarify the message of the paper.
Abstract
Line 14-16 (first sentence of the abstract): this sentence doesn’t read very well (‘odd sentence’), consider changing to: “The molecular protonation profiles obtained by means of an organic electrochemical transistor, which is used for analysis of molecular products released by blood-derived cultures, contain a large amount of information.”
Line 18: ‘the liquid biopsy’ is incorrect, please change to ‘liquid biopsies’ or ‘derived from liquid biopsy’
Line 18-20: ‘Odd sentence’. Consider changing to: “In the extracellular space of cultured cells the number of glycation products increase driven both by a glycolysis metabolism and by a compromised function of the glutathione redox system.”
Line 21-23: This sentence is unclear/partially incorrect. Consider changing “On this way” to “Due to this”, “to counteracting” = incorrect time match. “Interrelating the cancer risk”, consider rephrasing for clarity.
Line 24: please change “through” to “by”
Line 24-27: This sentence doesn’t read very well and is unclear. Consider changing the first part to something like “This study provides a novel chemometric approach for molecular analysis of protonation and discusses the possibility of constructing a predictive cancer risk model based on….
Introduction
Line 31-33: ‘Odd sentence’.
Line 34: please change ‘form’ to ‘from’.
Line 44-45: incorrect English.
Line 46: ‘play the’ = incorrect verb.
Line 61: ‘to counteracting’ wrong time match.
Line 70: remove word ‘based’.
Line 70-72: consider adding more depth by commenting on literature or other studies than your own that also prove this statement.
Line 75: “aims the possibility” is incorrect.
Material and Methods
2.1
Line 78: wrong time match
Data files S1 and S2 missing?
It would be of value to describe the patient group more elaborately, particularly some information on the diagnosis/main types of cancer.
2.5 Please improve the English.
Results
3.1
Line 206-214: English could be improved.
Figure 1.b /c small drops not really visible in pdf format?
3.2
Please improve the English.
Most of this part belongs in the methods section.
3.3-3.6
Quality of the figures is poor.
Discussion
Line 361: ‘based on experimental profiles collected on samples’ incorrect English.
Line 369-371: this sentence doesn’t read very well “odd sentence”.
First part of discussion until line 392 is more suitable for the methods section.
Line 401-405: incorrect English.
Line 407-409: this is a bold statement and also different for every tumour type or country. Where is the reference?
We have corrected all the points set by the reviewer as tracked in the revised manuscript and the paragraph 3.2 has been modified moving the details on patients and methods in the proper section.
We have re-written the discussion and the references were implemented.
We have also improved the quality of all figures
Reviewer 3 Report
The manuscript titled "Tailoring chemometric models on blood-derived cultures secretome to assess personalized cancer risk score" is, in my opinion, a well designed and structured research work exploring the use of new biochemical models to predict high risk of cancer (here named as intermediate). Indeed, subjects at high risk of cancer could be included in follow-up programs, as for those having a diagnosed genetic predisposition reducing then the costs of treating a cancer diagnosed in an advanced stage. Even if the small cohort of the study, the results here presented are encouraging and could be represent a valid tool for cancer management in the near future. Moreover, the methods used well fit with the aim of the research along with a proper statistical analysis. However, I would strongly advice for a careful read of the text for the presence of many typos along with language errors and sentences construction above all in results and discussion section.
Author Response
Paper: Manuscript ID: cancers-775627
Title: Tailoring chemometric models on blood-derived cultures secretome to assess personalized cancer risk score
Dear Editor,
We are pleased to submit to Cancers, a revised version of the manuscript entitled “Tailoring chemometric models on blood-derived cultures secretome to assess personalized cancer risk score”, and a detailed response to the comments of the Reviewers. The MS has been revised to comply with the observations of the Reviewers. Specifically:
o We have corrected typos everywhere in the MS and we improved English and style of the MS. Where necessary, we have rewritten words or sentences to convey more clearly the message of the paper.
o We have re-written the discussion.
o We have improved the quality of figures.
Considering the corrections made to the manuscript, and the positive comments of all the Reviewers on the work, we hope that the paper can be accepted for publication in this present form. We warmly thank the Reviewers that, with their comments, contributed to improve the work.
In the following, you will find the original comments from the reviewers (bold black text) and the point-by-point response of the authors (bold blue text). In the manuscript all modifications are tracked
With Regards
The authors
Review 3:
The manuscript titled "Tailoring chemometric models on blood-derived cultures secretome to assess personalized cancer risk score" is, in my opinion, a well designed and structured research work exploring the use of new biochemical models to predict high risk of cancer (here named as intermediate). Indeed, subjects at high risk of cancer could be included in follow-up programs, as for those having a diagnosed genetic predisposition reducing then the costs of treating a cancer diagnosed in an advanced stage. Even if the small cohort of the study, the results here presented are encouraging and could be represent a valid tool for cancer management in the near future. Moreover, the methods used well fit with the aim of the research along with a proper statistical analysis. However, I would strongly advice for a careful read of the text for the presence of many typos along with language errors and sentences construction above all in results and discussion section.
We have corrected typos, language errors and sentences’ construction in results and discussion sections. We thank the reviewer for the encouraged comment and we hope that the revised version of the MS will meet your precious expectations.
Round 2
Reviewer 2 Report
The authors made several revisions that clearly improved the quality of their paper